# Targeted Screening of *Lactiplantibacillus plantarum* Strains Isolated from Tomatoes and Its Application in Tomato Fermented Juice

**DOI:** 10.3390/foods13223569

**Published:** 2024-11-08

**Authors:** Nuersiman Tuerhong, Liang Wang, Jie Cui, Dilireba Shataer, Huizhen Yan, Xiaoxiao Dong, Ziqi Gao, Minwei Zhang, Yanan Qin, Jing Lu

**Affiliations:** 1Xinjiang Key Laboratory of Biological Resources and Genetic Engineering, College of Life Science and Technology, Xinjiang University, Urumqi 830046, China; 15099330343@163.com (N.T.); wl1390593786@163.com (L.W.); dlrb_shataer2021@xju.edu.cn (D.S.); 18536983054@163.com (H.Y.); 15599675517@163.com (X.D.); gaoziqi666@sina.com (Z.G.); zhangmw@xju.edu.cn (M.Z.); qingyalan12345@sina.com (Y.Q.); 2College of Food Science and Light Industry, Nanjing Tech University, Nanjing 211816, China; jiecui_njtech@163.com

**Keywords:** *Lactiplantibacillus plantarum*, tomato fermented juice, antioxidant activity, carbohydrate utilization, acid tolerance, antibiotic resistance, probiotic potential

## Abstract

This study explores the functional attributes of *Lactiplantibacillus plantarum* (*L. plantarum*) strains isolated from fermented tomato juice, focusing on their physiological, biochemical, and probiotic characteristics. The identified 66 gram-positive strains included 36 *L. plantarum* ones, which exhibited robust growth in acidic environments (pH 2.0–5.0) and utilization of various carbohydrates. Notably, seven strains outperformed a commercial strain in extreme acidic conditions. Antioxidant activity varied, with strain A24 showing the highest hydroxyl radical scavenging ability, while strains with high surface hydrophobicity had lower DPPH scavenging activity, indicating no direct correlation between these properties. Strains also showed strain-specific differences in carbohydrate utilization and antibiotic resistance, with some resistant to gentamicin and ciprofloxacin. Survival rates under simulated gastrointestinal conditions were strain-specific, with some strains demonstrating high survival rates, indicating their potential as probiotics. Furthermore, 13 strains used as fermentation starters in tomato juice significantly enhanced antioxidant activity and reduced pH and total soluble solids, indicating efficient sugar utilization and lactic acid production. These findings suggest that *L. plantarum* strains are well-suited for functional food fermentation and probiotic applications, with strain-specific traits offering versatility for use in acidic food products and probiotic formulations.

## 1. Introduction

*Lactiplantibacillus plantarum* (*L. plantarum*), a gram-positive, rod-shaped, non-spore-forming lactic acid bacterium (LAB) [1], exhibits robust ecological and metabolic adaptability and can be found in various environments, including fermented dairy products, fruits, and vegetables [2,3,4]. It is frequently employed as a starter culture for the fermentation of vegetables and fruits such as cherries, blueberries, mulberries, and pomegranates, enhancing the taste, texture, and sensory qualities of these food products [5,6,7].

Certain strains of *L. plantarum*, originating from fruits and vegetables, exhibit probiotic properties and the ability to thrive in plant-based food matrices, which can present more challenging conditions for probiotic survival compared to dairy products [8,9]. For instance, Cango et al. isolated indigenous LAB starters from tomatoes, including *L. plantarum*, *Lacticaseibacillus casei* (*L. casei*), and *Lactobacillus* sp. These strains, notably *L. plantarum*, enhanced the health-promoting, sensory, and shelf-life attributes of tomato juice compared to both unfermented tomato juice and tomato juice fermented with the autochthonous starter *L. plantarum* LP54, previously isolated from green olive fermentation brines [10].

Research has also shown that certain *L. plantarum* strains possess resilience in extreme environments. Nath et al. discovered that *L. plantarum* GCC19M1, isolated from fermented milk products, exhibited high resistance to low pH conditions [11]. When exposed to synthetic gastric juice at a pH of 3, this strain maintained survival rates between 93.48% and 96.97% [12]. In a separate study, 113 LAB strains were isolated from 24 spontaneously fermented chili samples, with *L. plantarum* GXL9 demonstrating the highest hydroxyl radical and 2,2-diphenyl-1-picrylhydrazyl (DPPH) scavenging activities [13]. Additional research indicated that fermenting tomato juice with *L. plantarum* and *L. casei* significantly increased the inhibitory levels of DPPH. Moreover, *L. plantarum* CCMA 0743, isolated from cauim (a traditional Brazilian beverage made from fermented cassava or maize), was retained with a count of 6.18 log CFU/mL following simulated digestion within a passion fruit juice matrix [14]. These findings underscore the strain-specific robustness of *L. plantarum*, rendering them ideal for probiotic applications across a range of food matrices.

*L. plantarum* demonstrates a versatile carbohydrate utilization profile, capable of metabolizing a variety of sugars such as cellobiose, mannose, D-ribose, L-fucose, fructo-oligosaccharides, and galacto-oligosaccharides, with the specific utilization patterns varying among strains depending on their origin. *L. plantarum* strains were isolated from tomatoes employing both phenotypic and genotypic methodologies. While the majority of these strains utilized N-acetyl-D-mannosamine and dextrin, *L. plantarum* POM 36 did not metabolize D-cellobiose or lactose. Cui et al. assessed the carbohydrate fermentation capabilities of *L. plantarum* strains LP-F1, LP-E1, LP-A4, LP-I5, LP1, and LP-4, isolated from fermented dairy samples. Their findings indicated that all strains were capable of utilizing D-glucose, D-fructose, D-mannose, D-ribose, D-mannitol, and D-sorbitol, but not galactitol, L-fucose, L-rhamnose, or D-trehalose. The strains also displayed differential fermentation efficiencies for D-mannitol, D-sorbitol, L-arabinose, and L-xylitol [15]. Mollova et al. investigated *L. plantarum* PU3 from human breast milk and discovered that this strain efficiently utilized a range of carbohydrates, including D-sorbitol, D-mannitol, and D-gluconic acid, as substrates for growth [16]. In summary, *L. plantarum* strains from various origins exhibit specific carbohydrate utilization profiles. While most research has concentrated on strains from fermented dairy products and human breast milk, there is a paucity of data on *L. plantarum* isolated from tomatoes. It is unclear whether tomato-derived strains differ in their capacity to metabolize monosaccharides, disaccharides, and polysaccharides or if they share similar carbohydrate utilization patterns.

*L. plantarum* EGER41, isolated from Kenyan spontaneously fermented milk, underwent antibiotic susceptibility testing, revealing sensitivity to azithromycin, tetracycline, and chloramphenicol, intermediate sensitivity to gentamicin, and resistance to nalidixic acid, ampicillin, and ciprofloxacin [17]. Shao et al. conducted a comparative study on antibiotic resistance among 17 *L. casei* strains from fermented cow’s milk or fermented yak’s milk, and 15 *L. plantarum* strains from sauerkraut and other fermented dairy products, found that *L. plantarum* and *L. casei* strains exhibit differing antibiotic resistance profiles [18]. Their findings indicated that all tested *L. plantarum* strains were susceptible to ampicillin, tetracycline, ciprofloxacin, and erythromycin, but resistant to vancomycin. Notably, *L. plantarum* typically exhibited higher minimum inhibitory concentration (MIC) values than *L. casei*, with all *L. plantarum* isolates being classified as resistant, particularly to aminoglycoside antibiotics. This highlights the diversity in antibiotic resistance patterns among various *L. plantarum* strains. Investigating the antibiotic susceptibility of *L. plantarum* is essential for its role as a starter culture in fruit and vegetable fermentation processes. Such studies ensure the safe utilization of *L. plantarum* in these fermentations, especially in settings where there may be concerns about residual antibiotics.

Furthermore, the probiotic characteristics of *L. plantarum* vary by strain, emphasizing the need to thoroughly characterize individual strains for tailored applications. For example, *L. plantarum* DMDL 9010, isolated from naturally fermented paocai, displayed advantageous traits such as adhesion ability, antioxidant activity, antibacterial effects, and gastrointestinal tolerance [19]. Similarly, in a study by Surve et al., *L. plantarum* DKL3 and JGR2, isolated from dhokla batter and jaggery, respectively, showed adhesion levels to human intestinal epithelial cells comparable to the well-known probiotic strain *Lacticaseibacillus rhamnosus* GG, with adhesion rates of 82.8% and 79.6%, respectively [20]. These results underscore the potential of *L. plantarum* strains from various origins to serve as effective probiotics with significant health benefits. Consequently, it is imperative to comprehend and investigate the unique characteristics of each *L. plantarum* strain to optimize its application in fermented food products and probiotic formulations, as these attributes directly impact its effectiveness in enhancing gut health and delivering other functional advantages.

This work aimed to isolate indigenous *L. plantarum* strains from tomatoes and assess their probiotic characteristics. This involved identifying *L. plantarum*, evaluating its physiological attributes, probiotic activity, safety profile, and carbon source utilization. Strains with potential probiotic benefits were selected and utilized in the production of probiotic fermented tomato juice. The outcomes of this study enhance our understanding of *L. plantarum* fermentation in tomato juice and provide guidance for the development of safe and nutritionally enriched fermented tomato enzyme products.

## 2. Materials and Methods

### 2.1. Preparation of Tomato Puree

Tomatoes (*Solanum lycopersicum* L.) from the 105th Regiment farmlands in Wujiaqu, Xinjiang, China, provided by Xinjiang Huizei Food Co., Ltd. (Urumqi, China), were processed under laboratory conditions. The Tunhe H1015 variety, characterized by an average fruit weight of 60 g, a solid content of 5.3%, a lycopene content of 13.1 mg/100 g, and a viscosity value of 12.5, was used. The tomatoes were juiced, and 227 g of concentrate was extracted, which was then thoroughly mixed with 723 mL of sterile water. After homogenizing the tomato puree, 100 mL of the liquid was transferred to an Erlenmeyer flask. This was heated in a water bath at 55 °C for 3 h to deactivate enzymes, followed by sterilization at 90 °C for 30 min. One aliquot of the tomato puree was supplemented with 2% glucose, while another was left without the addition of glucose.

The two variants of tomato puree are subsequently fermented at room temperature (23 °C) or at 37 °C for 16 h, resulting in four distinct batches of fermented tomato liquid. Throughout the fermentation process, the relative humidity is meticulously controlled to remain within the 60% to 80% range.

### 2.2. Isolation and Purification of Strains

A total of 25 mL of each tomato fermented puree is combined with 225 mL of sterilized deMan Rogosa Sharpe (MRS) medium containing 0.01% vancomycin. The strains were incubated at 37 °C for 16 h. Subsequently, the strains were serially diluted with sterile saline solution to achieve a final dilution of 1 × 10^7^ of the original concentration. A 0.1 mL aliquot of this dilution is plated onto MRS agar plates supplemented with bromothymol blue and incubated at 37 °C for 24 h. Upon the appearance of individual colonies, those exhibiting color changes are selected and streaked onto fresh MRS agar plates. This streaking process is repeated three times to ensure the purification of the strain.

### 2.3. Identification of Indigenous LAB via 16S rRNA Sequence Analysis

The strains were incubated in MRS medium at 37 °C for 18 h. Following incubation, DNA was extracted using a DNA extraction kit (Tiangen Biotechnology Co., Ltd., Beijing, China). The total DNA was then amplified via Polymerase Chain Reaction (PCR) employing the universal 27F forward primer (5′-AGTTTTGATCTGTCTCCAG-3′) and the universal 1492R reverse primer (5′-GGTACTTTTACGACT-3′) [21]. The resulting PCR products were sequenced by Shanghai Sanguang Biotechnology Co., Ltd. (Shanghai, China). The sequences obtained were aligned against the NCBI BLAST database for identification.

### 2.4. Physiological Characterization

#### 2.4.1. Growth Tolerance

The LAB strains were cultivated in MRS broth at 37 °C for 18 h. The pH of the MRS medium was adjusted to 2, 3, 4, and 5. The activated strains were then inoculated into the various pH medium at a rate of 2% (*v*/*v*) for 24 h at 37 °C. The optical density (OD_625_) of the fermentation broth was measured at 0, 16, and 24 h post-inoculation.

#### 2.4.2. Determination of Surface Hydrophobicity

The surface hydrophobicity of the bacterial cells was evaluated using microbial adherence to xylene hydrocarbon [22]. The strains were cultured at 37 °C for 18 h, followed by centrifugation at 4 °C and 8000 rpm for 10 min. The supernatant was discarded, and the bacterial pellet was resuspended. The cells were washed twice with sterile PBS buffer (pH 7.2) and then resuspended in sterilized 0.1 M KNO_3_ solution. The concentration of the bacterial suspension was adjusted to 10^7^~10^8^ CFU/mL, and the absorbance at 600 nm was measured to ensure it reached 0.5 ± 0.2 (A_0_).

To assess hydrophobicity, 3 mL of the bacterial suspension was mixed with 1 mL of xylene. The mixture was vortexed and shook for 3 min, followed by a 20 min settling period at room temperature. The aqueous phase was then carefully aspirated, and its absorbance was measured (A_1_). The surface hydrophobicity was calculated using the following formula.
(1)$$\mathrm{Surface}\;\mathrm{Hydrophobicity}\;(\%)=\frac{\left(A0-A1\right)}{A0}\;\times \;100\%$$
where A_0_ represents the OD_600_ of the suspension before mixing with xylene, and A_1_ is the OD_600_ of the aqueous phase after incubation.

#### 2.4.3. Autoaggregation Assay

The procedure for assessing autoaggregation was adapted from established protocols with necessary adjustments [23]. Freshly grown cultures were subjected to centrifugation at 8000 rpm for 10 min at 4 °C. The supernatant was removed. The pellet was washed twice with sterile PBS buffer and resuspended in sterile PBS to achieve a concentration of 10^−7^~10^−8^ CFU/mL, targeting an OD_600_ of 0.5 ± 0.2 (A_0_). Subsequently, 2 mL of the cell suspension was vortexed for 10 s and incubated at 37 °C for 2 h. Following incubation, 1 mL of the supernatant was carefully removed, and its absorbance at 600 nm (A_1_) was measured. The autoaggregation percentage was calculated using the following formula.
(2)$$\mathrm{Auto}\;\mathrm{aggregation}\;(\%)=(1-\frac{A1}{A0})\;\times \;100\%$$

#### 2.4.4. Antioxidant Capacity Assessment

##### Preparation of Cells and Intracellular Cell-Free Extracts

The strains were cultured in MRS broth at 37 °C for 18 h. The bacterial solution was then centrifuged at 8000× *g* for 10 min to obtain the cell pellet. The pellet was washed at pH 7.2 using sterilized PBS. After that, the complete cell suspension was again suspended in PBS, and the concentration was adjusted to 1 × 10^−9^ CFU/mL, resulting in a concentrated bacterial cell suspension suitable for further research or analysis [24].

##### DPPH Free Radical Scavenging Ability

The DPPH free radical scavenging ability of *L. plantarum* strains was assessed using a modified method based on a previously published protocol [25]. A freshly prepared DPPH solution (0.2 mM) was mixed with 0.5 mL of bacterial suspensions or PBS (as a control). After rapid mixing, the mixture was incubated in the dark for 30 min. Instead of bacterial suspensions, deionized water was used to prepare the blank samples. The reduction in absorbance at 517 nm was measured using a spectrophotometer to determine the amount of DPPH scavenged. The scavenging ability was calculated by comparing the sample absorbance to the blank absorbance and determining the percentage decrease in absorbance. This modified approach allowed for the evaluation of the DPPH free radical scavenging activity of bacterial suspensions, providing insights into their antioxidant properties.
(3)$$\mathrm{DPPH}\;\mathrm{radical}\;\mathrm{scavenging}\;(\%)=[1-\frac{A517(sample)}{A517(control)}]\;\times \;100\%$$

#### 2.4.5. Hydroxyl Radical Scavenging Capacity Evaluation

The hydroxyl radical scavenging capacity of LAB strains was evaluated using a modified technique as previously described [24]. The following components were rapidly mixed: 0.5 mL of PBS (pH 7.4), 0.5 mL of intact cells or cell-free extract, 1 mL of 2.5 mM FeSO_4_, and 0.5 mL of 2.5 mM 10-Phenanthroline. Then, 1 mL of 20 mM H_2_O_2_ was added and the liquid was left to sit at 37 °C for 90 min, to initiate the reaction. The ability to scavenge hydroxyl radicals was then assessed by measuring the increase in absorbance at 536 nm using a spectrophotometer. The scavenging ability was determined by comparing the sample absorbance to a control and calculating the percentage decrease in absorbance. This modified methodology enabled the assessment of the strains’ capacity to scavenge hydroxyl radicals, offering insights into their antioxidant characteristics.
(4)$$\mathrm{Hydroxyl}\;\mathrm{radical}\;\mathrm{scavenging}\;(\%)=[\frac{A536(sample)-A536(blank)}{A536(control)-A536(blank)}]\;\times \;100\%$$

### 2.5. Carbohydrate Utilization Characterization

#### 2.5.1. Preparation of Carbon Source Media

The carbohydrate solutions were prepared following established protocols [26]. Each carbohydrate, including sorbitol, mannitol, cellobiose, trehalose, fructose, and glucose (purchased from Shanghai Macklin Biochemical Co., Ltd. and Shanghai Lanjing Technology Development Co., Ltd., Shanghai, China), was dissolved in distilled water at a final concentration of 5% (*w*/*v*). The solutions were then filtered through Minisart filters with a 0.45 μm pore size.

#### 2.5.2. Growth Assessment on Carbohydrates

Adopting a modified version of the experimental method by McLaughlin et al. [27], bacterial strains were inoculated at a 2% rate into 5 mL of MRS medium and incubated at 37 °C for 18 h. After three generations of continuous subculture, bacteria in the late logarithmic phase of the third generation were collected, washed twice with physiological saline, and resuspended.

The bacterial suspension was then inoculated at a 1% (*v*/*v*) rate into various carbon source media (sorbitol, mannitol, cellobiose, trehalose, fructose, and glucose), with four replicates each, and incubated at 37 °C for 24 h in a 96-well plate. Growth was evaluated by measuring the optical density at 600 nm (OD_600_), with glucose-supplemented MRS media serving as positive controls and sugar-free MRS media as negative controls. Growth capability was determined based on OD_600_ readings.

#### 2.5.3. Qualitative Utilization on Carbohydrates

The strain was inoculated at a 2% rate into 5 mL of MRS medium and incubated at 37 °C for 18 h. Following three generations of subculture, bacteria in the late logarithmic phase of the third generation were collected, washed twice with physiological saline, and resuspended. The bacterial suspension was then inoculated at a 2% (*v*/*v*) rate into MRS media containing 1% (*w*/*v*) bromocresol purple and various carbon sources (sorbitol, mannitol, cellobiose, trehalose, fructose, and glucose) in a 96-well plate. The plate was incubated at 37 °C for 24 to 48 h, and changes in media color were observed. Utilized substrates turned yellow, while non-utilized ones remained purple [26,28]. Glucose-supplemented and unsupplemented MRS media were used as positive and negative controls, respectively.

### 2.6. Safety Evaluation of LAB Strains

The selected LAB strains were inoculated at a 2% rate into 5 mL of MRS medium and incubated at 37 °C for 18 h. Antibiotic susceptibility was assessed in accordance with EFSA guidelines, employing established methodologies [29]. Serial two-fold dilutions of four antibiotics (gentamicin, kanamycin, ampicillin, and tetracycline, purchased from Beijing Solarbio Science & Technology Co., Ltd., Beijing, China) were prepared, and 100 µL of each dilution was dispensed into the wells of 96-well plates. Bacterial suspensions were standardized to an optical density (OD) of 0.16–0.2 at 625 nm, equivalent to 3 × 10^8^ CFU/mL, and subsequently diluted 1000-fold. Then, 100 µL of these suspensions was added to each well. The plates were incubated at 28 °C for 48 h, and the minimum inhibitory concentration (MIC) for each strain was determined as the lowest antibiotic concentration that inhibited visible growth. MIC determinations were performed in triplicate.

### 2.7. Gastrointestinal Tolerance of LAB Strains

The gastrointestinal transit tolerance of the bacterial strains was evaluated using the method described by Maragkoudakis et al. [30]. In brief, the strains were cultured in MRS broth at 37 °C for 18 h. The bacterial cells were then harvested and sequentially resuspended in simulated gastric juice (0.5% *w*/*v* sterile saline with 3 g/L pepsin, pH 3.0, for 3 h) and simulated intestinal juice (0.5% *w*/*v* sterile saline with 1 g/L trypsin and 3 g/L bile salt, pH 8.0, for 4 h). Pepsin and trypsin were purchased from Beijing Solarbio Science & Technology Co., Ltd., Beijing, China and Shanghai Bicen Biochemical Technology Co., Ltd., Shanghai, China. Viable counts were determined via plate counting at 0, 3, and 7 h to assess the strains’ gastrointestinal transit tolerance. The specific viable bacterial counts at these time points reflect the gastrointestinal transit tolerance.
(5)Strain survival rate (%)=lg(N1 − N0) × 100%
where N1 represents the number of viable bacteria after treatment with simulated gastric or intestinal fluid, measured in CFU/mL; N0 represents the number of viable bacteria before treatment, also measured in CFU/mL.

### 2.8. Fermentation of Tomato Juice by Tomato-Derived L. plantarum

Following the removal of any atypical or diseased tomatoes, only red, ripe tomatoes were selected for juicing. The soluble solids were balanced with white sugar to achieve a Brix level of 12. The total soluble solids (Brix) were measured using a handheld Brix meter. Tomato juice naturally fermented without added strains served as a blank control, while values for samples inoculated with different strains were recorded after 22 h, with measurements taken in triplicate. Enzymatic hydrolysis was performed at 55 °C for 3 h using a mixture of 0.2% pectinase, 0.1% cellulase, and 0.1% hemicellulose (*v*/*v*) provided by Novozymes in Beijing, China. The mixture was then sterilized at 90 °C for 20 min and cooled to 37 °C in a sterile environment [31]. Tomato-derived strains (designated A33, A7, A17, A25, A3, A321, A27, A31, A2, A34, A5, A4, and A35) were introduced into the sterile tomato juice at a concentration of 4 × 10^7^ CFU/mL. Additionally, three generations of activated commercial strains—*L. plantarum* CICC25155 (ZW), *L. casei* CICC6114 (GL), and *Lactobacillus fermentum* CICC25124 (FJ)—purchased from the China Center of Industrial Culture Collection (Beijing, China) were used as control strains. Currently, Xinjiang Huize Food Co., Ltd. has utilized these three commercial strains in the industrial production of fermented tomato juice.

### 2.9. Analysis of Physicochemical Properties of Fermented Tomato Juice

SOD activity was determined following the manufacturer’s instructions (Nanjing Jiancheng Co., Ltd., Nanjing, China). To prepare fermented tomato juice, a 2 mL aliquot was centrifuged at 4 °C and 3000× *g* for 5 min. The supernatant was collected, diluted two-fold, and SOD activity was measured at OD_550_ nm. SOD activity (U/g) was calculated using the following formula:(6)$$SOD\left(U/g\right)=\frac{Contrast\;OD-Measurement\;OD}{Contrast\;OD}÷50\%\;\;\;\;\;\;\;\;\;\;\;\;\;\;\;\;\;\;\;\;\;\times \;Multiples÷Homogenizing\;Consistency\;(g/mL)$$

TSS was measured using a digital refractometer (model TD-45, Zhejiang, China) by pipetting a 1 mL drop of the sample into the saccharimeter injector and recording the reading. The pH was determined using a digital desktop acidimeter (model PHS-3C, Shanghai, China). Antioxidant capacity for enzyme-treated and control samples was evaluated using ABTS radical scavenging, according to the manufacturer’s guidelines (Nanjing Jiancheng Co., Ltd., Nanjing, China).

### 2.10. Statistical Analysis

All experimental data were collected in triplicate for consistency and analyzed using the mean or standard error. The ANOVA test of variance was used to assess the significance of intergroup differences in the aforementioned experimental data. The LSD letter labeling was used to indicate significance. Images were generated by using R4.2.2.

## 3. Results and Discussion

### 3.1. Identification and Physiological and Biochemical Characteristics Analysis of LAB Derived from Tomato Fermented Puree

*L. plantarum* exhibited significant tolerance to the fruit juice environment and robust fermentative capabilities, having been isolated from cherry tomatoes and various other fruits [32]. This species has also been previously identified in raw and spontaneously fermented fruits [33]. From the tomato fermented puree, a total of 66 gram-positive bacteria were identified, with 16S rDNA analysis revealing that 54% (36 strains) were *L. plantarum*. The strain tested negative for catalase and indole production and positive for gelatin liquefaction and did not produce carbon dioxide gas (Appendix A).

The biochemical test results for these strains show that the acid resistance of the 36 *L. plantarum* strains was evaluated by monitoring their growth variability at 0, 16, and 24 h in MRS broth with pH levels ranging from 2.0 to 5.0 (Figure 1). All *L. plantarum* strains displayed rapid growth from pH 2.0 to 5.0 and from 0 to 16 h, but growth slowed between 16 and 24 h. Notably, significant differences in survival were observed at 0, 16, and 24 h in acidic environments (pH 2.0 to pH 5.0), with an upward growth trend (*p* < 0.05). At pH 5.0 and pH 4.0, all 36 strains thrived. However, at pH 3.0 after 24 h, growth patterns varied, and the ability to grow diminished. Among these, eight strains outperformed the ZW strain in growth tolerance. As the pH decreased further to 2.0, viability significantly declined, with growth markedly inhibited. Only seven strains (designated A11, A25, A33, A34, A35, A36, and A37) demonstrated superior growth performance compared to the ZW strain under these extreme conditions.

Studies have indicated that the DPPH radical-scavenging activity of isolated strains is indicative of their antioxidant effects, serving as a key criterion for evaluating probiotic properties [34,35]. The antioxidant capacity and surface hydrophobicity of the *L. plantarum* strains are depicted in Figure 2, revealing strain-specific variations (*p* < 0.05). Out of the 36 strains evaluated, strain A24 exhibited the highest hydroxyl radical scavenging ability at 1 × 10^9^ CFU/mL. Strains A29, A7, and A30 also demonstrated robust activity. The DPPH radical scavenging activity of these strains is presented in Figure 2a, with strains ZW, FJ, A3, and A37 showing the highest activity, while strains A32, A33, A9, and GL exhibited the lowest. Interestingly, strains A31, A32, and A33 had the highest surface hydrophobicity (171.73%, 153.89%, and 137.40%) at 1 × 10^9^ CFU/mL, whereas strain A25 had the lowest. Notably, despite their high hydrophobicity, strains A31, A32, and A33 displayed the weakest DPPH scavenging ability (74.89%, 21.75%, and 33.31%). The antioxidant capacity, as measured via DPPH radical-scavenging activity, varied considerably among the strains. Strain A24 showed the highest hydroxyl radical scavenging ability, with strains A29, A7, and A30 also exhibiting strong antioxidant effects. Intriguingly, an inverse correlation was observed between surface hydrophobicity and antioxidant activity. Despite high hydrophobicity, strains A31, A32, and A33 had weaker DPPH scavenging activity. High hydrophobicity in these strains, likely due to cell wall proteins or lipids enhancing hydrophobic interactions, may divert resources from antioxidant production. Consequently, such strains may rely on alternative antioxidant pathways that are less effective for free radical scavenging, as measured via the DPPH assay.

This study also delved into the autoaggregation capacity of the strains, as well as the surface hydrophobic characteristics of the standard strains. The results, as shown in Figure 2b, reveal significant differences in surface adhesion ability and autoaggregation capacity among the strains (*p* < 0.05).

These findings suggest that surface hydrophobicity does not necessarily correlate with antioxidant capacity, highlighting the functional diversity within *L. plantarum*. The results underscore significant variability among the 36 strains in terms of acid and alkali tolerance, antioxidant capacity, and surface hydrophobicity. These insights emphasize the importance of selecting specific strains based on their functional properties, such as probiotic potential or antioxidant activity, for targeted applications.

### 3.2. Carbohydrate Utilization by L. plantarum Strains

*L. plantarum* demonstrates a robust capacity to metabolize a range of carbohydrates, which underpins its extensive adaptability across various environments with differing carbohydrate profiles [36,37,38]. In our study, we observed strain-specific variations in the fermentation of tested sugars by *L. plantarum* (*p* < 0.05). As illustrated in Figure 3, strains GL, A2, A3, A16, A27, A33, A34, A35, and A37 displayed significant differences in the utilization of monosaccharides (glucose vs. fructose), disaccharides (trehalose vs. cellobiose), and polysaccharides (mannitol vs. sorbitol), all showing robust growth (OD_600_ > 0.5). Strains ZW, A23, A24, and A25 exhibited significant differences in the utilization of monosaccharides and polysaccharides. Strains A8, A18, and A19 showed significant differences in the utilization of disaccharides and polysaccharides. In contrast, strains A5, A6, A13, A14, A17, and A32 did not display significant differences in the utilization of monosaccharides, disaccharides, and polysaccharides.

Studies have indicated that *L. plantarum* is capable of metabolizing a variety of carbohydrates, including cellobiose, mannose, D-ribose, and L-fucose, in addition to possessing a comprehensive carbohydrate utilization system [12,15]. It primarily utilizes glucose and fructose via the glycolytic pathway [12]. For cellobiose, *L. plantarum* first hydrolyzes it into glucose monomers before completing fermentation [39,40]. The efficiency of trehalose utilization varies among different strains, a finding that aligns with the results of this study. Moreover, while *L. plantarum* can metabolize mannitol and sorbitol, its efficiency is generally lower compared to monosaccharides and disaccharides, and this also differs among strains. In this study, certain strains showed a more effective utilization of mannitol and sorbitol, which may be associated with their strong utilization of monosaccharides, as observed with strain A25.

Previous research has demonstrated *L. plantarum*’s ability to utilize cellobiose, mannose, D-ribose, and L-fucose, further underscoring its extensive carbohydrate utilization system [15,41]. Our findings corroborate reports that *L. plantarum* primarily uses glucose and fructose through the glycolytic pathway, with strain-specific variations in the utilization efficiency of disaccharides such as trehalose. The observed differences in trehalose metabolism across strains suggest that trehalose fermentation efficiency could be a crucial factor to consider when selecting *L. plantarum* for specific industrial or probiotic applications.

In conclusion, the strain-specific variations in carbohydrate utilization by *L. plantarum* offer a foundation for choosing strains with desirable characteristics for targeted applications. Strains that demonstrate robust growth across a broad spectrum of sugars, such as A25 and others, could be prioritized for environments or products with complex sugar profiles. Further research into the genetic underpinnings of these metabolic capabilities could provide deeper insights into optimizing *L. plantarum* for tailored industrial and probiotic uses.

### 3.3. Antibiotic Resistance Profile of L. plantarum

Ensuring that *L. plantarum*, when used as a probiotic, does not pose adverse effects such as the spread of antibiotic resistance is crucial, thus necessitating a thorough safety evaluation. As depicted in Figure 4, the susceptibility of 36 *L. plantarum* strains to ampicillin, gentamicin, kanamycin, and tetracycline was assessed using the broth microdilution method, and all strains were found to be susceptible as per the European Food Safety Authority (EFSA, 2018) guidelines. However, variations in antibiotic susceptibility were noted among the strains. These susceptibility differences suggest that various *L. plantarum* strains exhibit significant variations in acquiring and expressing resistance genes, influenced by their genetic backgrounds, natural resistance mechanisms, and environmental selective pressures. For ampicillin, 21 strains were sensitive (MIC: 2 mg/mL), whereas the remainder exhibited resistance. For gentamicin, all strains except A36 and A31 were sensitive (MIC: 16 mg/mL). For tetracycline, 25 strains were sensitive (MIC: 32 mg/mL). In summary, strains A2, A3, A4, A7, A9, A17, A22, A25, A28, A29, A33, and A35 did not surpass the MIC thresholds for any of the four antibiotics tested. Nonetheless, several strains exhibited resistance to gentamicin, tetracycline, and kanamycin. The resistance of certain *L. plantarum* strains to tetracycline may be attributed to natural or intrinsic resistance mechanisms. This is a well-documented phenomenon across various microbial species and underscores the significance of comprehending the evolutionary pressures that contribute to these resistance patterns. Investigating the mechanisms behind this resistance is essential for developing probiotics with a minimized risk of adverse interactions within the human gut microbiome.

Nduko et al. reported that *L. plantarum* EGER41, isolated from spontaneously fermented milk, exhibited sensitivity to azithromycin, tetracycline, and chloramphenicol, and intermediate sensitivity to gentamicin [17]. In our study, some *L. plantarum* strains demonstrated intermediate sensitivity to gentamicin while being sensitive to ampicillin. The observed resistance to tetracycline in certain strains is likely due to natural or intrinsic resistance mechanisms. Although this study did not directly address the transferability of these resistances, it highlights the need for further investigation. Previous research suggests that the transfer of antibiotic resistance from *L. plantarum* to other bacterial species does not present a direct risk to human health. Nonetheless, continuous monitoring and research are essential to track any shifts in resistance patterns, particularly as probiotic use becomes increasingly common. Future studies should concentrate on both the phenotypic and genotypic resistance profiles of *L. plantarum* strains involved in food fermentations to ensure the safety and efficacy of these probiotics.

### 3.4. Gastrointestinal Tolerance of L. plantarum Strains

As a probiotic, *L. plantarum* must remain viable in food matrices and endure the harsh conditions of the gastrointestinal tract, including stomach acidity ranging from pH 1.0 to 3.0 and bile salt concentrations of approximately 0.3% in the small intestine, to ensure its beneficial effects post-ingestion. The gastrointestinal viability of *L. plantarum* is illustrated in Figure 5. The 36 strains exhibited varying responses to simulated gastric and intestinal conditions, indicating strain-specific differences in survival (*p* < 0.05).

As shown in Figure 5a, at an inoculation level of 1 × 10^8^ CFU/mL, strain A13 displayed the highest survival rates after exposure to simulated gastric and intestinal juices, with 90.72% survival after 3 h in gastric juice and 61.31% survival after 4 h in intestinal juice. Strains A3, A33, A27, A25, A35, and A17 also showed survival rates of 96.35%, 90.70%, 89.19%, 88.37%, 86.14%, and 71.67% in gastric juice and 87.67%, 81.39%, 76.67%, 76.05%, 67.54%, and 65.78% in intestinal juice, respectively. These strains maintained survival rates above 50% after exposure to both gastric and intestinal conditions. In contrast, strain GL and other strains exhibited lower survival rates (less than 30%) following intestinal exposure.

At an inoculation level of 1 × 10^9^ CFU/mL, as depicted in Figure 5b, strain A5 had the highest survival rates after both gastric (89.04%) and intestinal exposure (77.41%). Strains A4 and A7 also demonstrated survival rates of 77.37% and 74.63% in gastric juice, and 73.43% and 63.65% in intestinal juice, respectively. However, strains A31, A15, A8, and A6 had lower survival rates (less than 40%) after intestinal exposure.

Studies have shown that for *L. plantarum* to function effectively as a probiotic, it must contain a high number of viable bacterial cells and maintain a high survival rate through the gastrointestinal tract [42,43]. The survival rate of *L. plantarum* in artificial gastrointestinal juice is a crucial indicator of its potential probiotic efficacy [44]. A probiotic culture must endure the physical and chemical challenges of the gastrointestinal tract, such as acid, digestive enzymes, and bile acids [45].

These findings underscore the importance of selecting *L. plantarum* strains with robust survival capabilities for probiotic applications. As previous studies have shown, effective probiotic strains must not only have high initial viability but also withstand the acidic conditions of the stomach (pH 1.0–3.0), digestive enzymes, and bile salts (approximately 0.3%) in the small intestine [46,47]. The ability of certain *L. plantarum strains* in this study to maintain high survival rates under these conditions suggests they could be strong candidates for use as starter cultures in functional foods or dietary supplements aimed at promoting gut health.

The significant strain-specific differences in survival also underscore the need for targeted selection of *L. plantarum* based on their gastrointestinal resilience. Further studies into the genetic and molecular mechanisms contributing to these survival differences could provide deeper insights into optimizing these strains for commercial probiotic applications.

### 3.5. Effect of L. plantarum on the Antioxidant Activities and Physicochemical Properties of Fermented Tomato Juice

In this study, 13 strains of *L. plantarum* were used as single-strain fermentation starters to assess their influence on the antioxidant activities and physicochemical properties of fermented tomato juice (Figure 6a–d). Superoxide dismutase (SOD), a key enzyme with potent antioxidant properties, is produced during the fermentation of fruit extracts. LAB and yeasts can enhance antioxidant capacity by synthesizing enzymes such as SOD, glutathione peroxidase, NADH oxidase, and NADH peroxidase [48]. As shown in Figure 6a, significant variations in SOD activity were observed among the 13 strains compared to the commercial strains (FJ, GL, and ZW). Strain A33 exhibited the highest SOD activity, followed by strains A7 and A17. Strains A25, A3, A21, and A31 showed comparable SOD production, whereas strains A27, A2, A4, and A34 displayed progressively lower activities (Figure 6a).

Research has shown that fermentation of food can improve free radical scavenging abilities, although the mechanisms affecting antioxidant activity are complex [49,50]. The increase in ABTS radical scavenging activity suggests enhanced antioxidant capacity, likely due to release of active compounds during fermentation. As depicted in Figure 6b, ABTS radical scavenging activity decreased after fermentation with the 13 *L. plantarum* strains, with significant differences between strains compared to the control (*p* < 0.05). After 22 h, strains A27 and A35 achieved ABTS radical scavenging activities of 75.49% and 76.08%, respectively, while the other nine strains showed progressively lower activities. Our results indicate that, compared to non-commercial strains and tomato juice without added fermentation strains, the SOD and ABTS antioxidant activities of tomato juice significantly improved after 22 h fermentation with the 13 strains. This suggests that all 13 strains are suitable for tomato juice fermentation.

Under single-strain fermentation, the total soluble solids (TSS) content of the 13 strains showed significant differences and gradually decreased compared to the three commercial strains (FJ, GL, and ZW) (Figure 6c). Notably, after 22 h fermentation, some strains exhibited no significant difference in TSS content, indicating similar fermentation effects and efficiency. For example, there was no significant difference in TSS content between strains A2 and A33, and among strains A31, A4, A7, and A21.

As illustrated in Figure 6d, the fermentation of tomato juice with different strains, compared to three commercial strains at 22 h, resulted in a slight decrease in pH across all strains. The pH ranged from 3.27 to 3.94, with the lowest pH observed in Strain A5. It was suggested that α-tomatine might be released during the fermentation process, and its content could increase with rising acidity. This could explain the observed pH increase in fermented juices after several hours of fermentation. The application of *L. plantarum* to the fermentation of fruit and vegetable juices or pulps leads to a decrease in pH due to lactic acid production, which depends on the strain’s acid production capacity [51,52].

The 13 strains used in this study demonstrated similar fermentation and acid production capacities, indicating that all strains possess good fermentation and acid production abilities. These findings are consistent with previous results for fermented oat and barley beverages inoculated with *L. plantarum* and mulberry juice fermented with LAB [53,54].

The results of this study demonstrate that the 13 *L. plantarum* strains, when used as single-strain fermentation starters, significantly impacted the antioxidant activity and physicochemical properties of the tomato juice. Notably, strain A33 exhibited the highest SOD activity, followed by strains A7 and A17, indicating their strong potential for enhancing antioxidant properties during fermentation. The production of antioxidant enzymes like SOD is a critical factor in the scavenging of reactive oxygen species, contributing to the overall health benefits of fermented products. These findings are consistent with previous reports that fermentation enhances antioxidant capacity by releasing bioactive compounds through hydrolysis. However, the observed inter-strain differences in both SOD and ABTS radical scavenging activities suggest that the choice of strain is crucial for optimizing the antioxidant potential of the final product.

In addition to antioxidant activity, the 13 strains also influenced the TSS and pH of the fermented tomato enzyme. The gradual decrease in TSS content across all strains, with some strains showing no significant differences after 22 h fermentation, indicates efficient sugar utilization during fermentation. This decrease is consistent with typical lactic acid fermentation processes, where sugars are metabolized into lactic acid, leading to a reduction in TSS and a corresponding decrease in pH. The slight reduction in pH observed across all strains confirms their strong acid production capacity, which is critical for preserving fermented products and enhancing their sensory properties. These results align with findings from previous research that showed the fermentation process enhances antioxidant capacity by breaking down and releasing bioactive compounds through hydrolysis. For instance, *L. plantarum-fermented* tomato juice has been observed to increase the levels of phenolic compounds and other bioactives, contributing to improved radical scavenging capacity and overall health benefits [55].

Overall, the 13 *L. plantarum* strains demonstrated strong fermentation potential, antioxidant activity, and acid production capacity, making them suitable candidates for use in tomato enzyme fermentation. The variability in strain performance underscores the importance of selecting specific strains based on the desired functional properties, such as antioxidant enhancement or acid production, for optimized fermentation outcomes.

## 4. Conclusions

This study highlights the substantial influence of *L. plantarum* on the antioxidant activity and physicochemical properties of tomato juice during fermentation. The 13 strains selected for this research displayed notable strain-specific variations in antioxidant enzyme production, such as SOD, and in ABTS radical scavenging activity, emphasizing their potential to enhance the health benefits of fermented products. Strains A33, A7, and A17, in particular, exhibited the highest antioxidant activity, positioning them as promising candidates for applications in functional foods. Furthermore, these strains showcased robust fermentation capabilities, as evidenced by the progressive reduction in TSS and pH levels, which are critical indicators of fermentation efficiency and acid production. These findings underscore the importance of strain selection to tailor fermentation processes to achieve desired functional outcomes, such as improved antioxidant properties or increased acid production. In summary, the 13 *L. plantarum* strains evaluated in this study are well-suited for tomato juice fermentation and hold potential for application in the development of functional foods and beverages.

Future research could explore genetic and metabolic factors underlying strain-specific antioxidant activity, potentially identifying key genes or pathways that boost antioxidant enzyme production and radical scavenging. Additionally, optimizing fermentation conditions, such as temperature and time, could maximize health benefits of specific strains. Sensory analysis of tomato juice fermented with various *L. plantarum* strains would offer insights into consumer preferences and acceptance.

## Figures and Tables

**Figure 1 foods-13-03569-f001:**
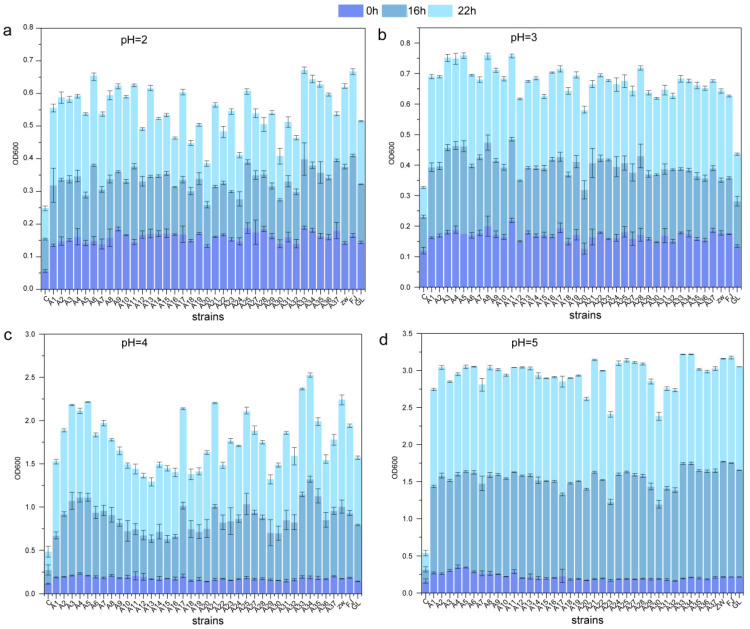
Growth tolerance of *L. plantarum* at various pH levels. Panels (**a**–**d**) illustrate the growth capabilities of *L. plantarum* at pH 2.0, 3.0, 4.0, and 5.0, respectively, under 0, 16, and 24 h. GL represents *L. casei* CICC6114; ZW represents *L. plantarum* CICC25155; FJ represents *L. fermentum* CICC25124.

**Figure 2 foods-13-03569-f002:**
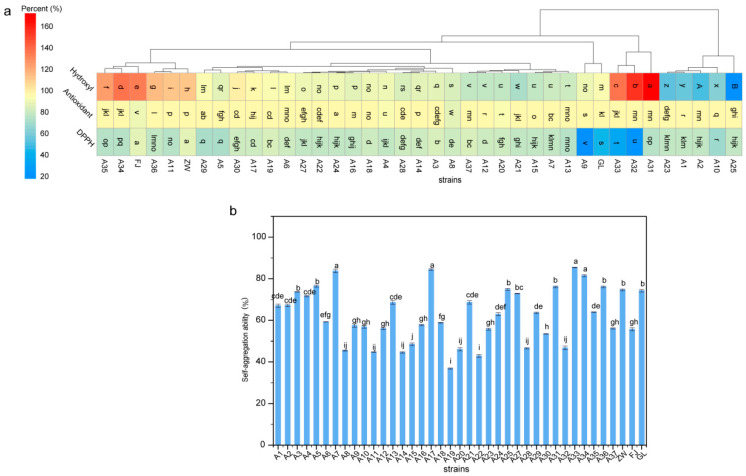
Antioxidant capacity and surface hydrophobicity of *L. plantarum strains*. (**a**) Illustrates the hydroxyl radical, antioxidant, and DPPH radical-scavenging activities of *L. plantarum strains*. (**b**) Depicts the autoaggregation capacity of *L. plantarum strains*. GL represents *L. casei* CICC6114; ZW represents *L. plantarum* CICC25155; FJ represents *L. fermentum* CICC25124. Different letters on the top indicate signficant in hyroxyl radical, antioxidant, DPPH and autoaggregation capacity of *L. plantarum strains* (*p* < 0.05, ANOVA Significant Difference test. The LSD letter labeling was used to indicate significance).

**Figure 3 foods-13-03569-f003:**
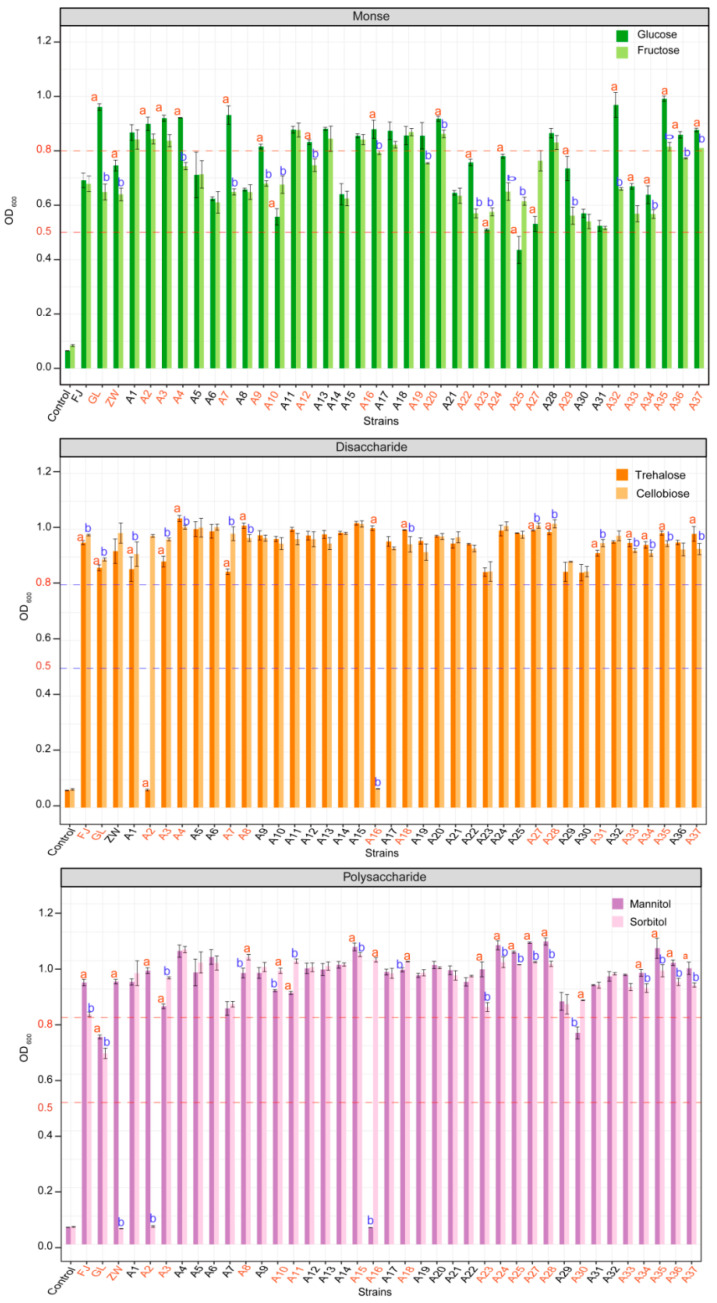
Carbohydrate utilization by *L. plantarum* strains. Red font and Blue font highlights significant differences in utilization between the two sugars compared. GL represents *L. casei* CICC6114; ZW represents *L. plantarum* CICC25155; FJ represents *L. fermentum* CICC25124. Different letters at the top indicate significant differences in Monosaccharide, Disaccharide, and Polysaccharide levels of *L. plantarum* strains (*p* < 0.05, ANOVA Significant Difference test). The LSD letter labeling was used to indicate significance.

**Figure 4 foods-13-03569-f004:**
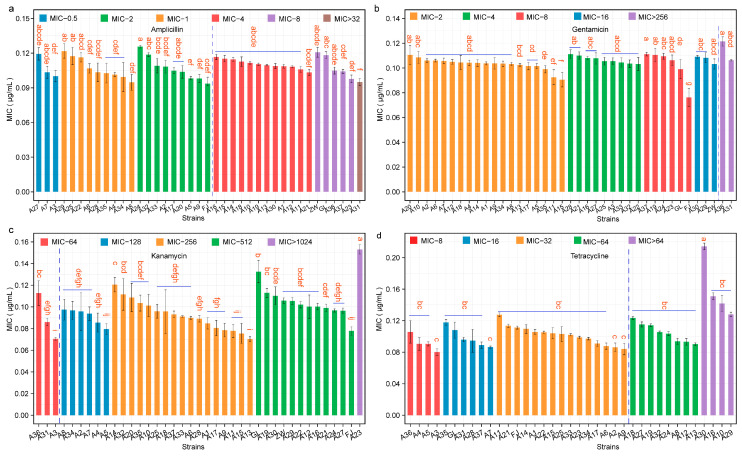
Antibiotic resistance profile of *L. plantarum* strains. Panels (**a**–**d**) correspond to resistance against ampicillin, gentamicin, kanamycin, and tetracycline, respectively. The dashed line indicates the antibiotic resistance threshold. GL represents *L. casei* CICC6114; ZW represents *L. plantarum* CICC25155; FJ represents *L. fermentum* CICC25124. Different letters at the top indicate significant differences in ampicillin, gentamicin, kanamycin and tetracycline of *L. plantarum* strains (*p* < 0.05, ANOVA Significant Difference test). The LSD letter labeling was used to indicate significance.

**Figure 5 foods-13-03569-f005:**
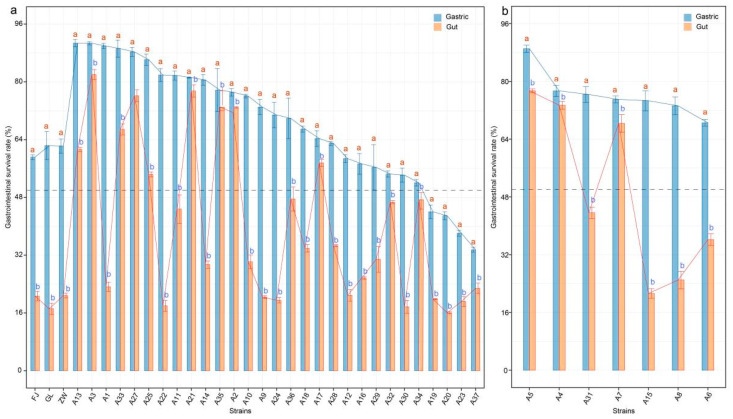
Gastrointestinal tolerance of *L. plantarum* strains. Panels (**a**,**b**) depict the gastrointestinal survival ability of *L. plantarum* evaluated at inoculation levels of 1 × 10^8^ CFU/mL and 1 × 10^9^ CFU/mL, respectively. Different letters above the bars indicate significant differences (*p* < 0.05, one-way ANOVA test). GL represents *L. casei* CICC6114; ZW represents *L. plantarum* CICC25155; FJ represents *L. fermentum* CICC25124.

**Figure 6 foods-13-03569-f006:**
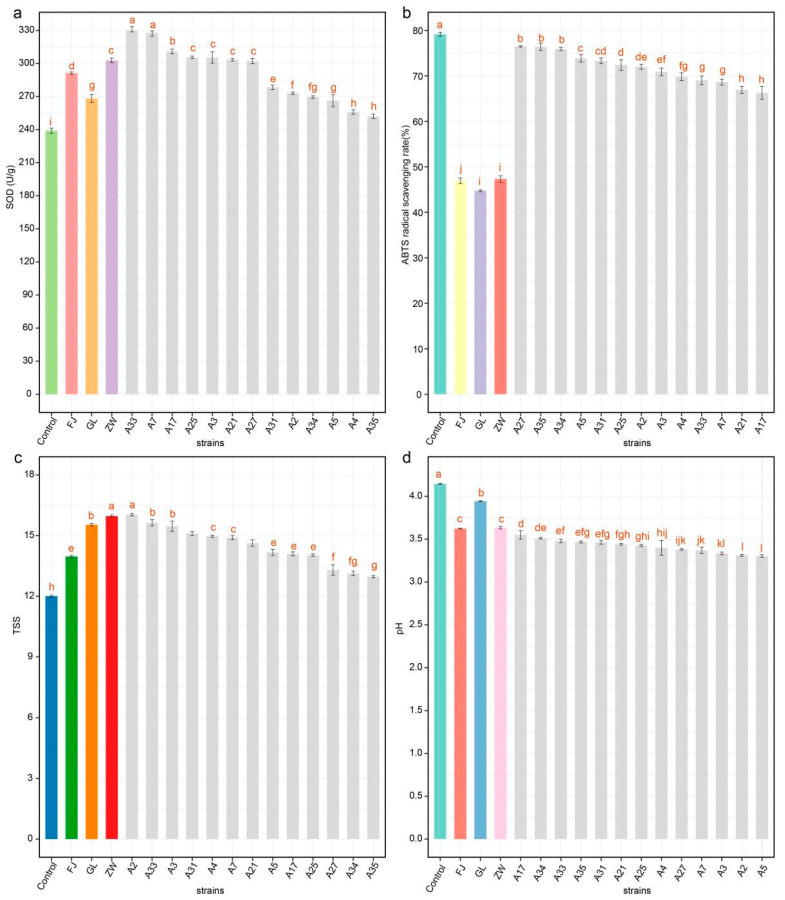
Antioxidant capacity and physiological characteristics of 13 *L. plantarum* strains in tomato juice fermentation. (**a**–**d**) show the SOD activity, ABTS scavenging capacity, TSS characteristics, and pH value of tomato juice fermentation after 22 h fermentation. Different letters above the bars indicate significant differences (*p* < 0.05, one-way ANOVA test). GL represents *L. casei* CICC6114; ZW represents *L. plantarum* CICC25155; FJ represents *L. fermentum* CICC25124.

## Data Availability

The original contributions presented in the study are included in the article/Appendix A, further inquiries can be directed to the corresponding authors.

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
