# Peer review of "Targeted Screening of Lactiplantibacillus plantarum Strains Isolated from Tomatoes and Its Application in Tomato Fermented Juice"

_foods, 2024, doi:10.3390/foods13223569_

Round 1

Reviewer 1 Report

Comments and Suggestions for Authors

The reviewed publication is interesting and brings new knowledge to the science of food technology. The background provides a sufficient introduction to the study, and the aim is clearly stated.

Materials and methods

Lines 120-124: The preparation of fermented tomatoes is described very sparingly. There is no information about where they were obtained? Where were they purchased? What variety was it? Was the relative humidity known during fermentation? This information may be added.

The research methodology has been described correctly, but I have noticed an inconsistency. In some descriptions, the authors provide the manufacturer of the reagent (line 195) but in most cases they do not. I suggest that this be standardized. 

Results and discussion

The description of the results is good, but I have some minor comments. The horizontal axis (strains) in the graphs is practically illegible (lines 294-295,  317, 345 and 394). I find the data in figure 3 (line 345) unreadable. Can this be visually improved somehow?

In Figure 6b and in the description below (lines 477-488) the authors describe the results of the ABTS scavenging activity test, but they did not perform such a test. In the methodology there is a DPPH test. Please check if this is not a typo? In conclusions the authors mention ABTS scavenging activity (line 541). Please check it out.

Literature

Please check if the literature has been prepared in accordance with the journal's requirements. Should the journal name be italicized and in full, and not abbreviated? What about the year of publication, shouldn't it be bold? Several sources lack the "doi" number. These are: 25 (lines 628-629), 33 (lines 649-650), 39 (lines 664-665), 48 (lines 690-691), 51 (lines 698-700), 53 (lines 704-705) , 55 (lines 708-709) and 56 (lines 710-711).

In summary, I believe that after taking into account the reviewer's comments, the article is worth publishing in a journal.

Author Response

Response to Reviewer 1 Comments

1. Summary

Thank you for your comments concerning our manuscript entitled “Targeted Screening of Lactiplantibacillus plantarum from Tomato Sources and Its Application in Tomato Fermented Enzymes” (Manuscript ID: foods-3263882). Those comments are all valuable and very helpful for revising and improving our paper, as well as the important guiding significance to our researches. We have carefully considered all comments from you and revised our manuscript accordingly. The manuscript has also been double-checked, and the typos and grammar errors we found have been corrected. In the following section, we summarize our responses to each comment from you.

2. Questions for General Evaluation

Reviewer’s Evaluation

Response and Revisions

Does the introduction provide sufficient background and include all relevant references?

Yes

Thank you for your positive comments and valuable suggestions to improve the quality of our manuscript.

Are all the cited references relevant to the research?

Yes

Thank you for your positive comments and valuable suggestions to improve the quality of our manuscript.

Is the research design appropriate?

Can be improved

We would like to thank you for your careful reading and helpful comments, and we have incorporated additions and revisions into the manuscript accordingly.

Are the results clearly presented?

Must be improved

We gratefully appreciate for your valuable comment. We have scrutinized and revised the sections that are inadequately organized or lack clarity in our manuscript.

Are the conclusions supported by the results?

Can be improved

We would like to thank you for your careful reading and helpful comments, and we have incorporated additions and revisions into the manuscript accordingly.

3. Point-by-point response to Comments and Suggestions for Authors

Comments 1: [Lines 120-124: The preparation of fermented tomatoes is described very sparingly. There is no information about where they were obtained? Where were they purchased? What variety was it? Was the relative humidity known during fermentation? This information may be added.]

Response 1:

[Tomatoes (Solanum lycopersicum L.) from the 105th Regiment farmlands in Wujiaqu, Xinjiang, China, supplied by Xinjiang Huizei Food Co., Ltd., were processed under laboratory conditions. The Tunhe H1015 variety, with an average fruit weight of 60 grams, has a solid content of 5.3%, lycopene content of 13.1 mg/100g, and a viscosity value of 12.5. Tomatoes are juiced, and 227g of concentrate is extracted, then mixed thoroughly with 723 mL of sterile water. After thoroughly mixing the above tomato puree, take 100 mL of the tomato liquid and place it in an Erlenmeyer flask. Heat it in a water bath at 55°C for 3 hours to inactivate enzymes, followed by sterilization at 90°C for 30 minutes. One portion of the tomato puree is supplemented with 2% glucose, while another portion is without the addition of 2% glucose.

Both portions of the tomato puree are then fermented at room temperature (23°C) or at 37°C for 16 h, yielding four distinct portions of tomato fermentation liquid. During the tomato paste fermentation process, the relative humidity is maintained within the range of 60% to 80%.] (Line 124-134,Line 135-138)

Thank you for your valuable suggestion. We have enhanced the purchasing information.

Comments 2: [The research methodology has been described correctly, but I have noticed an inconsistency. In some descriptions, the authors provide the manufacturer of the reagent (line 195) but in most cases they do not. I suggest that this be standardized.]

Response 2:

[Each carbohydrate, including sorbitol, mannitol, cellobiose, trehalose, fructose, and glucose (purchased from Shanghai Macklin Biochemical Co., Ltd. and Shanghai Lanjing Technology Development Co., Ltd., Shanghai, China), was dissolved in distilled water at a final concentration of 5% (w/v). (Line 220- 223)

Serial two-fold dilutions of four antibiotics (gentamicin, kanamycin, ampicillin, and tetracycline, purchased from Beijing Solarbio Science & Technology Co., Ltd, Beijing, China) were prepared, and 100 µL of each dilution was dispensed into the wells of 96-well plates. (Line 252- 253) 

Pepsin and trypsin were purchased from Beijing Solarbio Science & Technology Co., Ltd, Beijing, China and Shanghai Bicen Biochemical Technology Co., Ltd., Shanghai, China. (Line 267-269)

Purchased from the China Center of Industrial Culture Collection (Beijing, China), were used as control strains. (Line 288-289)

TSS was measured using a digital refractometer (model TD-45, Zhejiang, China) by pipetting 1 mL drop of sample into the saccharimeter injector and recording the reading. The pH was determined using a digital desktop acidimeter (model PHS-3C, Shanghai, China). Antioxidant capacity for enzyme-treated and control samples was evaluated using ABTS radical scavenging, according to the manufacturer’s guidelines (Nanjing Jiancheng Co., Ltd.). (Line 299-304)]

Thank you for pointing it out. We have corrected it.

Comments 3: [The description of the results is good, but I have some minor comments. The horizontal axis (strains) in the graphs is practically illegible (lines 294-295,  317, 345 and 394). I find the data in figure 3 (line 345) unreadable. Can this be visually improved somehow?]

Response 3:

Figure 1 (Line 332-334)

Figure 2 (Line 359-361)

Figure 3 (Line 387-389)

Figure 4 (Line 440-442)

Thank you for pointing it out. We have made these figures visually improved.

Comments 4: [In Figure 6b and in the description below (lines 477-488) the authors describe the results of the ABTS scavenging activity test, but they did not perform such a test. In the methodology there is a DPPH test. Please check if this is not a typo? In conclusions the authors mention ABTS scavenging activity (line 541). Please check it out.]

Response 4:

[2.8 Analysis of Physicochemical Properties of Fermented Tomato Juice

SOD activity was determined following the manufacturer’s instructions (Nanjing Jiancheng Co., Ltd., Nanjing, China). To prepare fermented tomato juice, a 2 mL aliquot was centrifuged at 4°C, 3000 g for 5 minutes. The supernatant was collected, diluted twofold, and SOD activity was measured at OD550 nm. SOD activity (U/g) was calculated using the formula:

% × Multiples Homogenizing Consistency (g/mL)                                                        

TSS was measured using a digital refractometer (model TD-45, Zhejiang, China) by pipetting 1 mL drop of sample into the saccharimeter injector and recording the reading. The pH was determined using a digital desktop acidimeter (model PHS-3C, Shanghai, China). Antioxidant capacity for enzyme-treated and control samples was evaluated using ABTS radical scavenging, according to the manufacturer’s guidelines (Nanjing Jiancheng Co., Ltd.)] (Line 292-304)

Thank you for pointing it out. We sincerely apologize for the inconvenience caused and have taken the necessary steps to rectify the issue.

Comments 5: [Please check if the literature has been prepared in accordance with the journal's requirements. Should the journal name be italicized and in full, and not abbreviated? What about the year of publication, shouldn't it be bold? Several sources lack the "doi" number. These are: 25 (lines 628-629), 33 (lines 649-650), 39 (lines 664-665), 48 (lines 690-691), 51 (lines 698-700), 53 (lines 704-705) , 55 (lines 708-709) and 56 (lines 710-711).]

Response 5:

7.  Tosukhowong, A.; Visessanguan, W.; Pumpuang, L.; Tepkasikul, P.; Panya, A.; Valyasevi, R. Biogenic amine formation in Nham, a Thai fermented sausage, and the reduction by commercial starter culture, Lactobacillus plantarum BCC 9546. Food Chemistry 2011, 129, 846-853.doi:10.1016/j.foodchem.2011.05.033.                                      (Line 637-639)

24.  Li, S.; Zhao, Y.; Zhang, L.; Zhang, X.; Huang, L.; Li, D.; Niu, C.; Yang, Z.; Wang, Q. Antioxidant activity of Lactobacillus plantarum strains isolated from traditional Chinese fermented foods. Food Chemistry 2012, 135, 1914-1919. doi:10.1016/j.foodchem.2012.06.048.                 (Line 681-683)

25.  Chen, P.; Zhang, Q.; Dang, H.; Liu, X.; Tian, F.; Zhao, J.; Chen, Y.; Zhang, H.; Chen, W. Screening for potential new probiotic based on probiotic properties and α-glucosidase inhibitory activity. Food Control 2014, 35, 65-72.doi: 10.1016/j.foodcont.2013.06.027.                         (Line 684-686)

27.  McLaughlin, J.L.; Chang, C.; Smith, D. Bench-top bioassays for the discovery of bioactive natural products: an update. Studies in natural products chemistry 1991, 9, 383-409.doi:10.1016/S0031-9422(98)00187-3.                                       (Line 690-691)

32.  Shi, F.; Wang, L.; Li, S. Enhancement in the physicochemical properties, antioxidant activity, volatile compounds, and non-volatile compounds of watermelon juices through Lactobacillus plantarum JHT78 fermentation. Food Chemistry 2023, 15,420, 136146.doi:10.1016/j.foodchem.2023.136146.  (Line 703-705)

33.  Garcia, C.; Guerin, M.; Souidi, K.; Remize, F. Lactic fermented fruit or vegetable juices: Past, present and future. Beverages 2020, 6, 8.doi:10.3390/beverages6010008.                     (Line 706-707)

39.  Cui, Y.; Wang, M.; Zheng, Y.; Miao, K.; Qu, X. The carbohydrate metabolism of Lactiplantibacillus plantarum. International Journal of Molecular Sciences 2021,15,22,24,13452.doi: 10.3390/ijms222413452.                                                        (Line 721-722)

41.  Li, K.; Wang, L.; Yu, D.; Yan, Z.; Liu, N.; Wu, A. Cellobiose inhibits the release of deoxynivalenol from transformed deoxynivalenol-3-glucoside from Lactiplantibacillus plantarum. Food Chem (Oxf) 2022,20,4, 100077.doi:https://doi.org/10.1016/j.fochms.2022.100077.                 (Line 726-728)

42.  Liang, T.; Jiang, T.; Liang, Z.; Zhang, N.; Dong, B.; Wu, Q.; Gu, B. Carbohydrate-active enzyme profiles of Lactiplantibacillus plantarum strain 84-3 contribute to flavor formation in fermented dairy and vegetable products. Food Chemistry X 2023,25,20, 101036.doi:10.1016/j.fochx.2023.101036.  (Line 729-731)

48. Lee, J.; Jo, J.; Wan, J.; Seo, H.; Han, S.-W.; Shin, Y.-J.; Kim, D.-H. In Vitro Evaluation of Probiotic Properties and Anti-Pathogenic Effects of Lactobacillus and Bifidobacterium Strains as Potential Probiotics. Foods 2024, 22,13,14, 2301.doi: 10.3390/foods13142301.                  (Line 747-749)

51. SarıtaÅŸ, S.; Portocarrero, A.C.M.; Miranda López, J.M.; Lombardo, M.; Koch, W.; Raposo, A.; El-Seedi, H.R.; de Brito Alves, J.L.; Esatbeyoglu, T.; Karav, S.; et al. The Impact of Fermentation on the Antioxidant Activity of Food Products. Molecules 2024,21,29,16,3941.doi: 10.3390/molecules29163941.                                                   (Line 756-758)

53. Hao, Y.; Li, J.; Wang, J.; Chen, Y. Mechanisms of Health Improvement by Lactiplantibacillus plantarum Based on Animal and Human Trials: A Review. Fermentation 2024, 10, 73.doi:10.3390/FERMENTATION10020073.                                     (Line 762-763)

55. MarchwiÅ„ska, K.; Gwiazdowska, D.; JuÅ›, K.; GluziÅ„ska, P.; Gwiazdowska, J.; Pawlak-LemaÅ„ska, K. Innovative Functional Lactic Acid Bacteria Fermented Oat Beverages with the Addition of Fruit Extracts and Lyophilisates. Applied Sciences 2023, 13, 12707.doi: 10.3390/APP132312707. (Line 766-768)

56. Guan, X.; Zhao, D.; Yu, T.; Liu, S.; Chen, S.; Huang, J.; Lai, G.; Lin, B.; Huang, J.; Lai, C.; et al. Phytochemical and Flavor Characteristics of Mulberry Juice Fermented with Lactiplantibacillus plantarum BXM2.Foods 2024,23,13,17,2648.doi: 10.3390/foods13172648.           (Line 769-771)]

Thank you for pointing it out. We have corrected and renewed the references.

Reviewer 2 Report

Comments and Suggestions for Authors

In Materials and methods section (page 3, line 121): The variety of the tomatoes used in this study must be described, as well their chemical composition.

In Materials and methods section (page 6, lines 258-259): How this Brix level was measured?    

In Materials and methods section (page 6, lines 264-266): How the control strains were chosen? Add this information.

In Results and discussion section (page 8, lines 315-316): It is mentioned that “Despite high hydrophobicity, strains A31, A32, and A33 had weaker DPPH scavenging activity.” Is there any explanation for this result?

In Results and discussion section (page 10, lines 381-382): How the variations in antibiotic susceptibility noted among the strains can be explained?

In Conclusions: It is good to add some information about future investigations related to this topic.

Author Response

Response to Reviewer 2 Comments

1. Summary

Thank you for your comments concerning our manuscript entitled “Targeted Screening of Lactiplantibacillus plantarum from Tomato Sources and Its Application in Tomato Fermented Enzymes” (Manuscript ID: foods-3263882). The title has been revised to "Targeted Screening of Lactiplantibacillus plantarum Strains Isolated from Tomatoes and Their Application in Tomato Fermented Juice" in accordance with the reviewers' recommendations. Those comments are all valuable and very helpful for revising and improving our paper, as well as the important guiding significance to our researches. We have carefully considered all comments from you and revised our manuscript accordingly. The manuscript has also been double-checked, and the typos and grammar errors we found have been corrected. In the following section, we summarize our responses to each comment from you.

2. Questions for General Evaluation

Reviewer’s Evaluation

Response and Revisions

Does the introduction provide sufficient background and include all relevant references?

Yes

Thank you for your positive comments and valuable suggestions to improve the quality of our manuscript.

Are all the cited references relevant to the research?

Can be improved

We would like to thank you for your careful reading and helpful comments, and we have incorporated additions and revisions into the manuscript accordingly.

Is the research design appropriate?

Can be improved

We would like to thank you for your careful reading and helpful comments, and we have incorporated additions and revisions into the manuscript accordingly.

Are the results clearly presented?

Can be improved

We gratefully appreciate for your valuable comment. We have scrutinized and revised the sections that are inadequately organized or lack clarity in our manuscript.

Are the conclusions supported by the results?

Can be improved

We would like to thank you for your careful reading and helpful comments, and we have incorporated additions and revisions into the manuscript accordingly.

3. Point-by-point response to Comments and Suggestions for Authors

Comments 1: In Materials and methods section (page 3, line 121): The variety of the tomatoes used in this study must be described, as well their chemical composition.]

Response 1:

[Tomatoes (Solanum lycopersicum L.) from the 105th Regiment farmlands in Wujiaqu, Xinjiang, China, provided by Xinjiang Huizei Food Co., Ltd., were processed under labor-atory conditions. The Tunhe H1015 variety, characterized by an average fruit weight of 60 grams, a solid content of 5.3%, a lycopene content of 13.1 mg/100 g, and a viscosity value of 12.5, was used.] (Line 125-129)

Thank you for your valuable suggestion. We have added related information.

Comments 2: [In Materials and methods section (page 6, lines 258-259): How this Brix level was measured?]

Response 2:

[The total soluble solids (Brix) were measured using a handheld Brix meter. Tomato juice naturally fermented without added strains served as a blank control, while values for samples inoculated with different strains were recorded after 22 h, with measurements taken in triplicate.] (Line 278-281)

Thank you for pointing it out. We have added it.

Comments 3: [In Materials and methods section (page 6, lines 264-266): How the control strains were chosen? Add this information.]

Thank you for pointing it out. We have added related information.

Comments 4: [In Results and discussion section (page 8, lines 315-316): It is mentioned that “Despite high hydrophobicity, strains A31, A32, and A33 had weaker DPPH scavenging activity.” Is there any explanation for this result?]

Response 4:

[High hydrophobicity in these strains, likely due to cell wall proteins or lipids enhancing hydrophobic interactions, may divert resources from antioxidant production. Consequently, such strains may rely on alternative antioxidant pathways that are less effective for free radical scavenging, as measured by the DPPH assay.] (Line 355-359)

Thank you for pointing it out. We have added related information.

Comments 5: [In Results and discussion section (page 10, lines 381-382): How the variations in antibiotic susceptibility noted among the strains can be explained?]

Response 5:

[However, variations in antibiotic susceptibility were noted among the strains. These susceptibility differences suggest that various L. plantarum strains exhibit significant variations in acquiring and expressing resistance genes, influenced by their genetic backgrounds, natural resistance mechanisms, and environmental selective pressures.] (Line 424-428)

Thank you for pointing it out. We have added related information.

Comments 6: [In Conclusions: It is good to add some information about future investigations related to this topic.]

Response 6:

[Future research could explore genetic and metabolic factors underlying strain-specific antioxidant activity, potentially identifying key genes or pathways that boost antioxidant enzyme production and radical scavenging. Additionally, optimizing fermentation conditions, such as temperature and time, could maximize health benefits of specific strains. Sensory analysis of tomato juice fermented with various L. plantarum strains would offer insights into consumer preferences and acceptance.] (Line 602-607)

Thank you for pointing it out. We have added related information.

Reviewer 3 Report

Comments and Suggestions for Authors

The manuscript describes a study aimed at isolating and characterizing strains of L. plantarum from tomatoes and testing them as starters in the fermentation of tomato juice. The idea is valid in the sense of searching for potentially probiotic strains of that species to increase the functionality of foods. The organization of the work is adequate. However, there are several confusions in the text that make it difficult to read and review. The English needs several corrections. There are conceptual, methodological and writing problems, which are detailed below.

- first of all, there is a confusion that persists throughout the text, even in the title. The meaning and importance that the authors give to the word "enzymes" is not understood. There is talk of "tomato fermented enzymes", "tomato enzyme production", "tomato protease products", "tomato ferments", "properties of tomato enzyme", "tomato enzyme fermentation"......According to what is understood, the aim is to ferment tomato juice with wild strains of L. plantarum and see the increase in antioxidant activities , etc, in the product. I repeat, the enzymatic component must be eliminated when talking about the process unless it is well explained because this word is always included.

-  Title: the authors did not investigate a genus and species of lactic acid bacteria but investigated strains of a genus and species. Therefore, the title should be changed to "Targeted screening of Lactiplantibacillus plantarum strains isolated from tomatoes and its application for fermenting tomato products (juices)". Review the entire text and add, where necessary, the word "strains" when only genus and species are mentioned.

- 2.1: Explain the tomato sampling further. How many different samples were used? What was the size or weight of each sample?, etc.

- line 129: 1:107 ?

line 210: Explain what mMRS is.

- 2.6: Determining the gastrointestinal tolerance of strains is not synonymous with "probiotic character". To ensure this, much more is needed. Since 2.6 only talks about gastrointestinal transit tolerance, the title of this point should be: "Gastrointestinal tolerance of LAB strains"

- 2.7: ...vegetable enzyme production ? Explain well the environmental conditions in which the tomato juice was fermented with the strains. (time, temperature....)

- 3.1: ....from tomatoes (delete "ferments"). Also line 279

- 3.5: ...properties of tomato enzyme ?? Idem line 462

- Fig. 6: ...L. plantarum strains in Tomato....

              Tomato enzyme fermentation ?

Comments on the Quality of English Language

- Abstract, lines 15-16: "...(L. plantarum) strains isolated from fermented tomatoes, focusing on their physiological..... included 36 L. plantarum ones, which..."

- line 17: tolerance to various carbohydrates ? o utilization of various carbohydrates?

- Always write the genus and species of a microorganism in italics. Review References and the text

- line 135: The strains were incubated....

- Fig. 1: delete "conditions" (segunda linea)

- Fig. 2:  improve the title: "...of L. plantarum strains. (a) illustrates......activities of L. plantarum strains. (b) despicts the.....L. plantarum strains. GL represents...."

- 3.2, improve the title: "Carbohydrate utilization by L. plantarum strains"

- Fig. 3: "Carbohydrate utilization by L. plantarum strains. Red font..." Idem for Figs. 4 and 5

- 3.4: "Gastrointestinal tolerance of L. plantarum strains"

- line 449: ...certain L. plantarum strains in this...

Author Response

Response to Reviewer 3 Comments

1. Summary

Thank you for your comments concerning our manuscript entitled “Targeted Screening of Lactiplantibacillus plantarum from Tomato Sources and Its Application in Tomato Fermented Enzymes” (Manuscript ID: foods-3263882). The title has been revised to "Targeted Screening of Lactiplantibacillus plantarum Strains Isolated from Tomatoes and Their Application in Tomato Fermented Juice" in accordance with the reviewers' recommendations. Those comments are all valuable and very helpful for revising and improving our paper, as well as the important guiding significance to our researches. We have carefully considered all comments from you and revised our manuscript accordingly. The manuscript has also been double-checked, and the typos and grammar errors we found have been corrected. In the following section, we summarize our responses to each comment from you.

2. Questions for General Evaluation

Reviewer’s Evaluation

Response and Revisions

Does the introduction provide sufficient background and include all relevant references?

Yes

Thank you for your positive comments and valuable suggestions to improve the quality of our manuscript.

Are all the cited references relevant to the research?

Can be improved

We gratefully appreciate for your valuable comment. We have scrutinized and revised the sections that are inadequately organized or lack clarity in our manuscript.

Is the research design appropriate?

Can be improved

We would like to thank you for your careful reading and helpful comments, and we have incorporated additions and revisions into the manuscript accordingly.

Are the results clearly presented?

Can be improved

We gratefully appreciate for your valuable comment. We have scrutinized and revised the sections that are inadequately organized or lack clarity in our manuscript.

Are the conclusions supported by the results?

Can be improved

We would like to thank you for your careful reading and helpful comments, and we have incorporated additions and revisions into the manuscript accordingly.

3. Point-by-point response to Comments and Suggestions for Authors

Comments 1: [first of all, there is a confusion that persists throughout the text, even in the title. The meaning and importance that the authors give to the word "enzymes" is not understood. There is talk of "tomato fermented enzymes", "tomato enzyme production", "tomato protease products", "tomato ferments", "properties of tomato enzyme", "tomato enzyme fermentation"......According to what is understood, the aim is to ferment tomato juice with wild strains of L. plantarum and see the increase in antioxidant activities , etc, in the product. I repeat, the enzymatic component must be eliminated when talking about the process unless it is well explained because this word is always included.]

Response 1:

[Targeted Screening of Lactiplantibacillus plantarum strains isolated from Tomatoes and Its Application in Tomato Fermented Juice] (Line 2-4)

Thank you for your valuable suggestion. We have corrected it. The title has been revised to "Targeted Screening of Lactiplantibacillus plantarum Strains Isolated from Tomatoes and Their Application in Tomato Fermented Juice"

Comments 2: [Title: the authors did not investigate a genus and species of lactic acid bacteria but investigated strains of a genus and species. Therefore, the title should be changed to "Targeted screening of Lactiplantibacillus plantarum strains isolated from tomatoes and its application for fermenting tomato products (juices)". Review the entire text and add, where necessary, the word "strains" when only genus and species are mentioned.]

Response 2:

[Targeted Screening of Lactiplantibacillus plantarum strains isolated from Tomatoes and Its Application in Tomato Fermented Juice] (Line 2-4)

We sincerely thank you for your valuable comments that we have renewed and replenished this issue.

Comments 3: [2.1: Explain the tomato sampling further. How many different samples were used? What was the size or weight of each sample?, etc. line 129: 1:107 ? line 210: Explain what mMRS is. ]

Response 3:

[Tomatoes (Solanum lycopersicum L.) from the 105th Regiment farmlands in Wujiaqu, Xinjiang, China, provided by Xinjiang Huizei Food Co., Ltd., were processed under labor-atory conditions. The Tunhe H1015 variety, characterized by an average fruit weight of 60 grams, a solid content of 5.3%, a lycopene content of 13.1 mg/100 g, and a viscosity value of 12.5, was used. The tomatoes were juiced, and 227 g of concentrate was extracted, which was then thoroughly mixed with 723 mL of sterile water. After homogenizing the tomato puree, 100 mL of the liquid was transferred to an Erlenmeyer flask. (Line 125-132)

Subsequently, the strains were serially diluted with sterile saline solution to achieve a final dilution of 1×107 of the original concentration. (Line 142-143)

Adopting a modified version of the experimental method by McLaughlin et al. [27] bacterial strains were inoculated at a 2% rate into 5 mL of MRS medium and incubated at 37°C for 18 h. After three generations of continuous subculture, bacteria in the late logarithmic phase of the third generation were collected, washed twice with physiological saline, and resuspended.

The bacterial suspension was then inoculated at a 1% (v/v) rate into various carbon source media (sorbitol, mannitol, cellobiose, trehalose, fructose, and glucose), with four replicates each, and incubated at 37°C for 24 h in a 96-well plate. Growth was evaluated by measuring the optical density at 600 nm (OD600), with glucose-supplemented MRS media serving as positive controls and sugar-free MRS media as negative controls. Growth capability was determined based on OD600 readings. (Line 226-236)]

Thank you for pointing it out. “mMRS” is a typographical error. We have corrected it.

Comments 4: [2.6: Determining the gastrointestinal tolerance of strains is not synonymous with "probiotic character". To ensure this, much more is needed. Since 2.6 only talks about gastrointestinal transit tolerance, the title of this point should be: "Gastrointestinal tolerance of LAB strains"]

Response 4:

[2.6. Gastrointestinal tolerance of LAB strains ] (Line 261)

Thank you for pointing it out. We have corrected it.

Comments 5: [2.7: ...vegetable enzyme production ? Explain well the environmental conditions in which the tomato juice was fermented with the strains. (time, temperature....)]

Response 5:

[2.7 Fermentation of Tomato Juice by Tomato-Derived L. plantarum] (Line 275)

Thank you for pointing it out. We have corrected and added related information.

Comments 6: [3.1: ....from tomatoes (delete "ferments"). Also line 279]

Response 6

[3.1. Identification and Physiological and Biochemical Characteristics analysis of LAB derived from Tomato Puree  (Line 312-313)

From the tomato fermented puree, a total of 66 Gram-positive bacteria were identified, with 16S rDNA analysis revealing that 54% (36 strains) were L. plantarum. (Line 317-319)]

Thank you for pointing it out. We have corrected it.

Comments 7: [3.5: ...properties of tomato enzyme ?? Idem line 462]

Response 7:

[3.5. Effect of L. plantarum on the Antioxidant Activities and Physicochemical Properties of Fermented tomato juice ] (Line 506-507)

Thank you for pointing it out. We have corrected and added related information.

Comments 8: [Fig. 6: ...L. plantarum strains in Tomato....Tomato enzyme fermentation ?]

Response 8:

[Figure 6. Antioxidant Capacity and Physiological Characteristics of 13 L. plantarum strains in Tomato Juice Fermentation] (Line 520-521)

Thank you for pointing it out. We have corrected it.

Comments 9: [Abstract, lines 15-16: "...(L. plantarum) strains isolated from fermented tomatoes, focusing on their physiological..... included 36 L. plantarum ones, which..."]

Response 9

[This study explores the functional attributes of Lactiplantibacillus plantarum (L. plantarum) strains isolated from fermented tomatoes juice, focusing on their physiological, biochemical, and probiotic characteristics.] (Line 15-16)

Thank you for pointing it out. We have corrected it.

Comments 10: [line 17: tolerance to various carbohydrates ? o utilization of various carbohydrates?]

Response 10:

[The identified 66 Gram-positive strains included 36 L. plantarum ones, which exhibited robust growth in acidic environments (pH 2.0-5.0) and utilization of various carbohydrates.] (Line 17-18)

Thank you for pointing it out. We have corrected it.

Comments 11: [Always write the genus and species of a microorganism in italics. Review References and the text]

Response 11:

Thank you for pointing it out. We have corrected it.

Comments 12: [line 135: The strains were incubated...]

Response 12

[The strains were incubated at 37°C for 16 h. Subsequently, the strains were serially diluted with sterile saline solution to achieve a final dilution of 1×107 of the original concentration.] (Line 140-143 )

Thank you for pointing it out. We have corrected it.

Comments 13: [Fig. 1: delete "conditions" (segunda linea)]

Response 13:

[Figure 1. Growth Tolerance of L. plantarum at Various pH Levels.] (Line 334)

Thank you for pointing it out. We have corrected it.

Comments 14: [Fig. 2:  improve the title: "...of L. plantarum strains. (a) illustrates......activities of L. plantarum strains. (b) despicts the.....L. plantarum strains. GL represents...."]

Response 14:

[Figure 2. Antioxidant Capacity and Surface Hydrophobicity of L. plantarum strains. (a) illustrates the hydroxyl radical, antioxidant, and DPPH radical-scavenging activities of L. plantarum strains. (b) depicts the autoaggregation capacity of L. plantarum strains. GL represents L. casei CICC6114; ZW represents L. plantarum CICC25155; FJ represents L. fermentum CICC25124] (Line 361-364)

Thank you for pointing it out. We have corrected and added related information.

Comments 15: [3.2, improve the title: "Carbohydrate utilization by L. plantarum strains"]

Response 15

[3.2. Carbohydrate utilization by L. plantarum strains] (Line 375)

Thank you for pointing it out. We have corrected it.

Comments 16: [Fig. 3: "Carbohydrate utilization by L. plantarum strains. Red font..." Idem for Figs. 4 and 5]

Response 16:

[Figure 3. Carbohydrate utilization by L. plantarum strains. Red font highlights significant differences in utilization between the two sugars compared. GL represents L. casei CICC6114; ZW represents L. plantarum CICC25155; FJ represents L. fermentum CICC25124. (Line 389-391)

Figure 4. Antibiotic Resistance Profile of L. plantarum strains. Panels a-d correspond to resistance against ampicillin, gentamicin, kanamycin, and tetracycline, respectively. The dashed line indicates the antibiotic resistance threshold. GL represents L. casei CICC6114; ZW represents L. plantarum CICC25155; FJ represents L. fermentum CICC25124. (Line 442-445)

Figure 5. Gastrointestinal Tolerance of L. plantarum strains. Panels a-b depict the gastrointestinal survival ability of L. plantarum evaluated at inoculation levels of 1×108 CFU/mL and 1×109 CFU/mL, respectively. Different letters above the bars indicate significant differences (p < 0.05, one-way ANOVA test). GL represents L. casei CICC6114; ZW represents L. plantarum CICC25155; FJ represents L. fermentum CICC25124. (Line 468-472)]

Thank you for pointing it out. We have corrected and added related information.

Comments 17: [3.4: "Gastrointestinal tolerance of L. plantarum strains")]

Response 17:

[3.4. Gastrointestinal tolerance of L. plantarum strains] (Line 460)

Thank you for pointing it out. We have corrected and added related information.

Comments 18: [line 449: ...certain L. plantarum strains in this...]

Response 18:

[Studies have certain that for L. plantarum to function effectively as a probiotic, it must contain a high number of viable bacterial cells and maintain a high survival rate through the gastrointestinal tract] (Line 487-489)

Thank you for pointing it out. We have corrected and added related information.

Reviewer 4 Report

Comments and Suggestions for Authors

 The authors dealt with the isolation of indigenous L. plantarum strains from tomatoes and assessed their probiotic characteristics. The work included identifying L. plantarum, evaluating its physiological attributes, probiotic activity, safety profile, and carbon source utilization. Strains with potential probiotic benefits were selected and utilized in the production of probiotic fermented tomato enzymes. An interesting publication, consistent with the latest research trends. The research in the work was properly planned and carried out. The selection of research methods does not raise any objections. The results were collected and presented in graphic form. The results were subjected to statistical analysis, which makes it very easy to analyze such a large number of results. However, as a reviewer, it is necessary to mention the poorly visible standard deviations in Fig. 2, which makes it difficult to correctly assess the statistics contained in the graphs. In addition, Fig. 3 contains a lot of information that is not legible, while Fig. 4 does not contain a comparison of significance between individual samples. The presentation of the results does not raise any objections, as does the summary. The authors correctly selected and cited literature that is within the scope of research presented in the publication.

Author Response

Response to Reviewer 4 Comments

1. Summary

Thank you for your comments concerning our manuscript entitled “Targeted Screening of Lactiplantibacillus plantarum from Tomato Sources and Its Application in Tomato Fermented Enzymes” (Manuscript ID: foods-3263882). The title has been revised to "Targeted Screening of Lactiplantibacillus plantarum Strains Isolated from Tomatoes and Their Application in Tomato Fermented Juice" in accordance with the reviewers' recommendations. Those comments are all valuable and very helpful for revising and improving our paper, as well as the important guiding significance to our researches. We have carefully considered all comments from you and revised our manuscript accordingly. The manuscript has also been double-checked, and the typos and grammar errors we found have been corrected. In the following section, we summarize our responses to each comment from you.

2. Questions for General Evaluation

Reviewer’s Evaluation

Response and Revisions

Does the introduction provide sufficient background and include all relevant references?

Yes

Thank you for your positive comments and valuable suggestions to improve the quality of our manuscript.

Are all the cited references relevant to the research?

Yes

Thank you for your positive comments and valuable suggestions to improve the quality of our manuscript.

Is the research design appropriate?

Yes

Thank you for your positive comments and valuable suggestions to improve the quality of our manuscript.

Are the results clearly presented?

Can be improved

We gratefully appreciate for your valuable comment. We have scrutinized and revised the sections that are inadequately organized or lack clarity in our manuscript.

Are the conclusions supported by the results?

Yes

Thank you for your positive comments and valuable suggestions to improve the quality of our manuscript.

3. Point-by-point response to Comments and Suggestions for Authors

Comments 1: [However, as a reviewer, it is necessary to mention the poorly visible standard deviations in Fig. 2, which makes it difficult to correctly assess the statistics contained in the graphs.]

Response 1:

[ Figure 2. ( Line 359)

]

Thank you for your valuable suggestion. We have corrected it.

Comments 2: [In addition, Fig. 3 contains a lot of information that is not legible, while Fig. 4 does not contain a comparison of significance between individual samples.]

Response 2:

[Figure 3 ( Line 387)

Figure 4. (Line 441)

]

We sincerely thank you for your valuable comments that we have renewed and replenished this issue.
